# Recent Advances, Challenges, and Metabolic Engineering Strategies in L-Cysteine Biosynthesis

**Wenwei Li [1], Zhen Zhou [2] and Dan Wang [1],***

[1]  Department of Chemical Engineering, School of Chemistry and Chemical Engineering, Chongqing University, Chongqing 401331, China; wenweili189@163.com

[2]  School of Pharmaceutical Sciences, Chongqing University, Chongqing 401331, China; dominizhou@126.com

*   Correspondence: dwang@cqu.edu.cn; Tel.: +86-23-65678926

**Abstract:** L-Cysteine is a widely used unique sulfur-containing amino acid with wide application in the food, pharmaceutical, and agricultural industries. This paper concludes the advantages and disadvantages of chemical hydrolysis, enzymatic biotransformation, and fermentation for the synthesis of L-cysteine. Meanwhile, a detailed introduction is given to the biosynthesis of L-cysteine, metabolic engineering strategies, and the latest progress in reported L-cysteine fermentation bacteria. Finally, insights are provided on the development direction of increasing the production of biosynthetic L-cysteine in the future. This review provides ideas for the future development of more efficient L-cysteine biosynthetic pathways.

**Keywords:** L-cysteine; *Escherichia coli*; *Corynebacterium glutamicum*; metabolic engineering; synthetic biology

## 1. Introduction

L-cysteine is a sulfur-containing amino acid that plays an important role in the folding of proteins, has a high redox activity in cellular metabolism, is a catalytic residue for a variety of enzymes, and is a sulfur donor compound that is required for the synthesis of Fe/S clusters, biotin, coenzyme A, and thiamine [1–3]. In addition to its roles in cellular metabolism, L-cysteine plays a variety of roles in metal binding, catalytic activity, and redox and has a vast array of industrial applications in the production of food, cosmetics, pharmaceuticals, and animal feed [4–6].

Chemical hydrolysis of proteins, which are typically extracted from the keratin of animal hair such as feathers, pig hair, etc., is the traditional method of industrial production of L-cysteine [7]. However, this method not only consumes a large amount of hydrochloric acid, but it also causes an unpleasant odor and wastewater treatment problems, which have a significant impact on the environment [7]. To avoid the environmental hazards of this method, scientists have explored biotechnological approaches to synthesize L-cysteine as an alternative to chemical hydrolysis. Fermentation and enzymatic biotransformation are the two most prominent biotechnological methods [8,9]. However, due to the presence of generated L-cysteine products that inhibit the activity of the enzymes, the enzyme bioconversion method presents difficulty in solving the problems of low yield and high cost. On the other hand, the fermentation method has the advantages of safety, low cost, and green and sustainable production, and as a result, it is the method that is currently considered to be the mainstream method [3,8,9]. The advantages and disadvantages of chemical hydrolysis, fermentation, and enzymatic biotransformation are summarized in Table 1.

Although fermentation offers a number of advantages, the design and construction of efficient microbial cell factories for fermentative production of L-cysteine remains challenging due to the high toxicity of L-cysteine and the complex regulation of its synthetic pathway [10]. The efficient production of L-cysteine on an industrial scale has not yet been

achieved, which is a major challenge for the industrialization of L-cysteine [11,12]. Numerous microorganisms, including bacteria such as *E. coli*, *C. glutamicum*, and *Pantoea ananatis*, have been engineered to produce L-cysteine due to the rapid development of systems metabolic engineering and synthetic biology [13–17]. In comparison to other bacteria, *E. coli* has a rapid growth rate and more developed genetic engineering techniques, whereas *C. glutamicum* is a non-pathogenic, industrial microorganism with developed fermentation technology that is extensively employed in food processing and other industries [13,18]. Therefore, *E. coli* and *C. glutamicum* are the two most studied chassis cells that directly produce L-cysteine from glucose [19–21].

This article compares the conventional chemical hydrolysis of keratin for L-cysteine extraction with the use of biotechnology for L-cysteine manufacture and describes current advancements in L-cysteine biosynthesis. The study also evaluates the merits and downsides of each method. In addition, the metabolic pathways of L-cysteine in *E. coli* and *C. glutamicum* are analyzed and summarized. Lastly, the metabolic strategies of L-cysteine production are comprehensively summarized, and the future bioproduction of L-cysteine is discussed.

**Table 1.** Comparison of Cysteine Production by Various Methods.

| Method | Raw Material | Bacteria | Advantage | Disadvantage | References |
|--------|-------------|----------|-----------|--------------|------------|
| Extraction from protein hydrolysates | Human hair and animal feathers | / | Raw materials are cheap and readily available. | Low yield, bad smell, wastewater treatment, and other problems. | [7] |
| Enzymatic transformation | DL-2-amino-$\Delta^2$-thiazoline-4-carboxylic acid (ATC) | *Pseudomonas* | Low energy requirements and environmental friendliness. | The activity of the enzyme is unstable, the yield is low, and the cost is relatively high. | [8,9] |
| Microbial fermentation | Glucose | *E. coli* or *C. glutamicum* or other bacteria | Raw materials are readily available, safe, low cost, green, and sustainable. | At present, the low yield is not suitable for industrial production. | [19–21] |

## 2. Advances in the Biosynthesis of L-Cysteine

### 2.1. Enzyme Biotransformation—Asymmetrical Hydrolysis of DL-2-amino-$\Delta^2$-thiazoline-4-Carboxylic Acid

Since L-cysteine is traditionally obtained by hydrolyzing animal hair, the extraction of 1 kg of L-cysteine requires approximately 10 kg of animal hair and 2.7 kg of hydrochloric acid, a process that not only has a low yield but also produces foul odors and wastewater, causing severe environmental damage [22]. The transformation method uses *Pseudomonas* to enzymatically convert DL-2-amino-$\Delta^2$-thiazoline-4-carboxylic acid (DL-ATC) to L-cysteine. This method of converting DL-ATC to cysteine involves three enzymes: ATC racemase, L-ATC hydrolase, and S-carbamoyl-L-cysteine hydrolase [23,24]. The complete procedure consists of three stages (Figure 1): (i) conversion of D-ATC to L-ATC by ATC racemase, (ii) ring-opening of L-ATC by L-ATC hydrolase to produce N-carbamoyl-L-cysteine (L-NCC), and (iii) final hydrolysis of L-NCC to L-cysteine by S-carbamoyl-L-cysteine hydrolase [25].

**Figure 1.** A metabolic pathway of DL-2-amino-$\Delta^2$-thiazoline-4-carboxylic acid (DL-ATC) to L-cysteine via *N*-carbamyl-L-cysteine (L-NCC) in *Pseudomonas* species. *atcB* gene encoding L-ATC acid hydrolase; *atcC* gene encoding L-NCC amidohydrolase.

Both L-ATC hydrolase (*atcB*) and S-carbamoyl-L-cysteine hydrolase (*atcC*) genes originated from *Pseudomonas* sp. strain BS. After sequencing by Japanese scientists, the amino acid sequence of the *atcC* gene product was found to be highly homologous to L-*N*-carbamoylase from other bacteria, but the amino acid sequence of the *atcB* gene was novel [8]. *AtcB* was initially identified as a gene encoding an enzyme that catalyzes the thiazoline ring-opening reaction and does not share a high degree of homology with previously described enzymes [26].

Enzymatic bioconversion is to some extent environmentally friendly and has lower energy consumption than hydrolysis of animal hair. However, the high toxicity of L-cysteine inhibits enzyme activity, leading to low efficiency and relatively high cost [8,9].

### 2.2. Biological Fermentation Methods

### 2.2.1. L-Cysteine Biosynthesis in *E. coli*

It is well known that in most microorganisms and plants, L-serine is the precursor substance for the synthesis of L-cysteine. The biosynthetic pathway of L-cysteine has been widely reported after many years of research [4,27,28]. In gut bacteria, L-serine is synthesized via a three-step pathway from the glycolytic intermediate 3-phosphoglycerate, and L-cysteine is synthesized via a two-step pathway from L-serine [28]. Firstly, the glycolytic intermediate 3-phosphoglycerate is converted to L-serine by a three-step reaction catalyzed by 3-phosphoglycerate dehydrogenase (PGDH), phosphoserine aminotransferase (PAST), and phosphoserine phosphatase (PSP) (Figure 2) [27]. L-serine is then further converted to L-cysteine by the catalytic action of serine *O*-acetyltransferase (CysE) encoded by the *cysE* gene and L-cysteine synthase (CysK) encoded by the *cysK* gene [27]. Due to the cellular toxicity of L-cysteine, the L-cysteine biosynthetic pathway is tightly regulated by several systems, including feedback inhibition of enzymes, degradation, and cellular efflux, resulting in a blocked biosynthetic pathway [29,30].

Scientists have conducted studies to address the aforementioned causes of L-cysteine biosynthesis blockage. Table 2 summarizes the research progress on the fermentation synthesis of L-cysteine by different engineering strains. Firstly, relevant research has been conducted on the feedback inhibition of the biosynthetic pathway of L-cysteine. In 1986, Denk et al. isolated an *E. coli* mutant that can excrete L-cysteine [31]. As a consequence of their investigation, they discovered that a mutation in *cysE*, the structural gene for serine CysE in *E. coli*, reduced the enzyme's sensitivity to L-cysteine. To investigate the structural basis for cysteine excretion, the *cysE* genes of mutant and wild-type *E. coli* were cloned, and their nucleotide sequences were determined. It was discovered that replacing the methionine residue at position 256 of the CysE polypeptide with an isoleucine residue decreased the feedback sensitivity of L-cysteine by roughly tenfold [31]. This modification allowed the L-cysteine efflux into the culture medium to increase to approximately 0.03 g/L. By modifying the *cysE* gene, it is possible to construct *E. coli* that produces high levels of L-cysteine, as indicated by the above results. Therefore, Nakamori et al. used site-directed mutagenesis to replace the methionine residue at position 256 in SAT with other amino acids or to terminate codons by truncating the carboxyl ends of amino acid residues at positions 256 to 273. The modified *cysE* gene was then expressed and constructed in *E. coli* to produce high levels of L-cysteine from glucose, increasing L-cysteine productivity to 0.20 g/L [32].

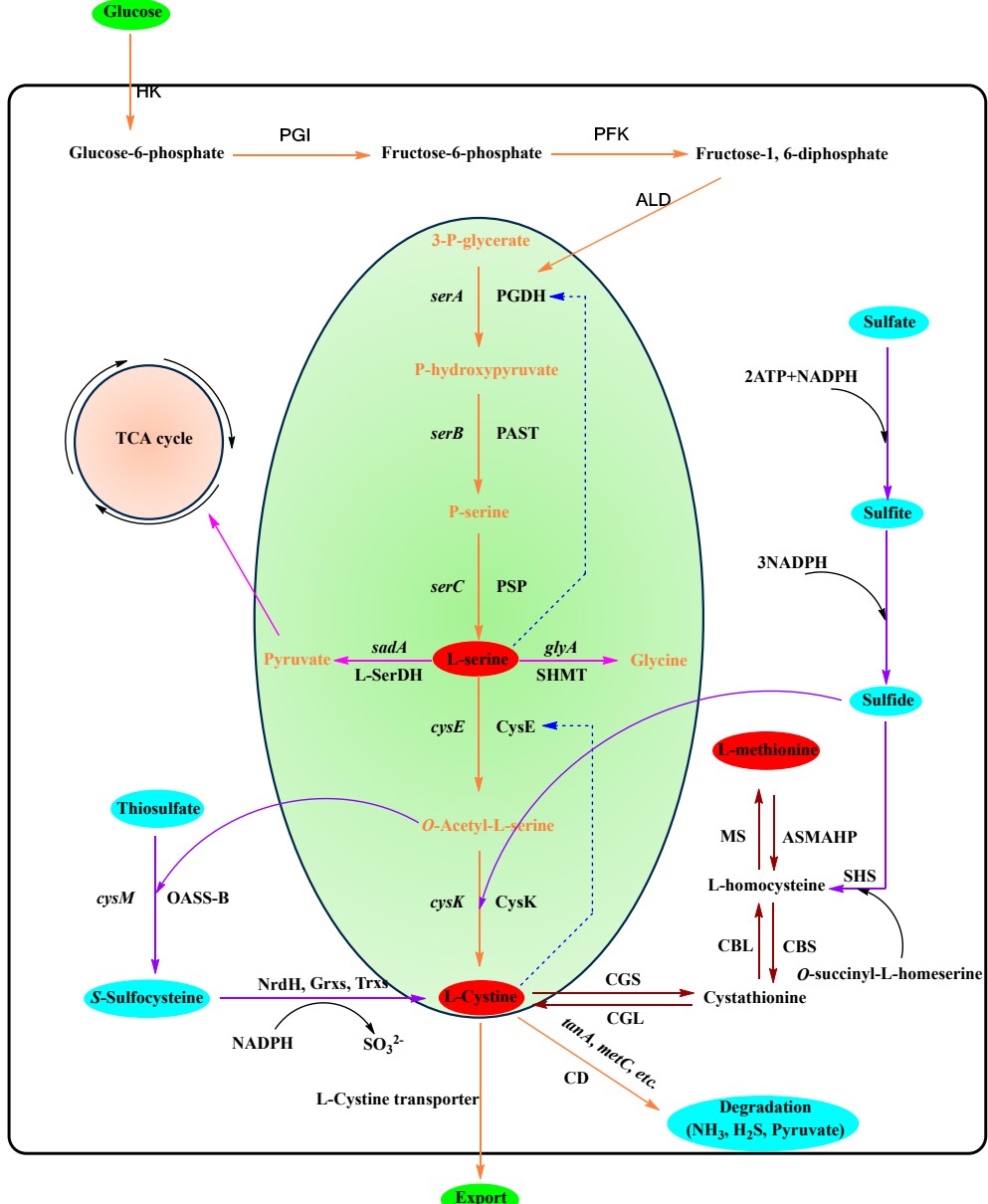

**Figure 2.** The metabolic pathway of L-cysteine in *E. coli* and *C. glutamicum*. Brown arrows refer to metabolic pathways in *P. aeruginosa*. The solid purple line indicates the conversion of L-serine to other productsDashed lines represent feedback inhibition. The solid orange line indicates the major metabolic pathway of L-cysteine. The solid purple line indicates the source of the sulfide. Italicized fonts on the same arrow line indicate genes encoding corresponding enzymes. HK, hexokinase; PGI, phosphoglucose isomerase; PFK, phosphofructokinase; ALD, aldolase; PGDH, 3-phosphoglycerate dehydrogenase; PAST, phosphoserine aminotransferase; PSP, phosphoserine phosphatase; CysE, serine *O*-acetyltransferase; CysK, cysteine synthase; L-SerDH, L-serine dehydratase; SHMT, serine hydroxymethyl transferase; CD, L-cysteine desulfhydrases; OASS-B, *O*-acetyl-L-serine sulfhydrylase-A; NrdH and Grxs, glutaredoxins; Trxs, thioredoxins; CGS, cystathionine γ-synthase; CBL, cystathionine β-lyase; MS, methionine synthase; SHS, O-succinyl-L-homoserine sulfhydrylase; ASMAHP, S-adenosylmethionine synthase-methyltransferases-S-adenosylhomocysteine hydrolase pathway; CBS, cystathionine β-synthase; CGL, cystathionine γ-lyase.

**Table 2.** Progress of fermentation synthesis of L-cysteine by different engineered strains.

| Bacterial Strain | Metabolic Strategy | L-Cysteine Production (g/L) | Productivity (g/(L·h)) | References |
|---|---|---|---|---|
| *E. coli* JM240 | Enhancing biosynthesis | 0.03 | / | [31] |
| *E. coli* JM39 | Enhancing biosynthesis | 0.20 | 0.003 | [32] |
| *E. coli* W3110 | Enhancing excretion | 0.07 | 0.003 | [33] |
| *E. coli* W3110 | Enhancing excretion | 0.15 | 0.007 | [34] |
| *E. coli* JM39 | Enhancing biosynthesis and weakening degradation | 0.60 | 0.013 | [35] |
| *E. coli* MG1655 | Enhancing biosynthesis and excretion and weakening degradation | 1.20 | 0.025 | [30] |
| *E. coli* BW25113 | Enhancing biosynthesis and excretion | 1.23 | 0.026 | [36] |
| *E. coli* BW25113 | Enhancing biosynthesis and excretion/weakening degradation | 1.72 | 0.024 | [20] |
| *E. coli* JM109 | Enhancing the sulfur conversion rate | 7.50 | 0.341 | [14] |
| *E. coli* BW25113 | Enhancing biosynthesis and thiosulfate assimilation and weakening degradation | 8.34 | 0.321 | [10] |
| *E. coli* W3110 | Balancing carbon and sulfur module conversion rate | 11.94 | 0.254 | [37] |
| *C. glutamicum* IR33 | Enhancing biosynthesis | 0.29 | 0.004 | [38] |
| *C. glutamicum* ATCC13032; *C. glutamicum* ATCC21586 | Enhanced sulfur metabolism in biosynthesis | 0.06 | 0.004 | [27] |
| *C. glutamicum* NBRC12168 | Enhancing biosynthesis and weakening degradation | 0.20 | 0.017 | [16] |
| *C. glutamicum* CYS | Enhancing biosynthesis and excretion | 0.28 | 0.014 | [19] |
| *C. glutamicum* ATCC13032 | Enhancing precursor accumulation and weakening degradation | 0.95 | 0.026 | [15] |
| *C. glutamicum* Cys -10 | Enhancing biosynthesis, excretion, and sulfur metabolism and weakening degradation | 5.92 | 0.082 | [39] |
| *Pantoea ananatis* | Weakening degradation and educing efflux | 2.20 | 0.079 | [17] |

Secondly, research has been conducted on the issue of L-cysteine excretion in cells. For example, Böck et al. discovered the YdeD protein from *E. coli* and investigated the structural and functional properties of the YdeD protein, which was found to be associated with the efflux of metabolites from the L-cysteine pathway [33]. The structural and functional characterization of YdeD was elucidated in *E. coli*, which when overexpressed leads to high levels of excretion of several metabolites of the cysteine pathway. In a subsequent study, Böck et al. identified a new protein, Yfik, from *E. coli* that was more efficient than YdeD for the export of *O*-acetylserine (OAS) or L-cysteine, allowing the L-cysteine yield to be increased to 0.15 g/L [34]. In a recently published work, a novel EamB mutant with higher activity toward L-cysteine was obtained by directed evolution. Two favorable mutants, G156S and N157S, were found to have higher L-cysteine output compared to wild-type EamB. By combining advantageous sites, the optimal mutant N157S/G156S increased L-cysteine production by approximately 70% during shake-flask fermentation [12].

Thirdly, research has been conducted on the degradation of L-cysteine in cells. Since the prerequisite substance for the synthesis of L-cysteine is L-serine, it is possible to increase the yield of L-cysteine by reducing the degradation activity of L-serine and L-cysteine. In *E. coli*, the degradation of L-cysteine is mainly catalyzed by L-cysteine desulfurase (CD), and the degradation of L-serine is mainly catalyzed by tryptophanase (TNase) [40,41]. To date, the enzymes with CD activity in *E. coli* have been studied in considerable detail, and it was determined that five enzymes in *E. coli*, namely TNase, cystathionine β-lyase, *O*-acetylserine sulfhydrylase-A, *O*-acetylserine sulfhydrylase-B, and the MalY protein, have an effect on CD enzyme activity [35,41]. However, CD activity is mainly derived from two enzymes: TNase, encoded by the *tnaA* gene, and cystathionine β-lyase, encoded by the *metC* gene. *TnaA* and *metC* mutants transformed with a plasmid containing a feedback-insensitive serine *O*-acetyltransferase gene had higher L-cysteine yields than those of the wild-type strains carrying the same plasmid, reaching 0.60 g/L [35].

It has been found that, in solving the problems faced by several of the above L-cysteines individually to increase the yield of L-cysteine, although the yield is increased, it is not very effective and is still far from the goal of industrial production [32,34,35]. In recent years, with the rapid development of metabolic engineering and synthetic biology, various metabolic engineering strategies have been used in combination to significantly increase the yield of L-cysteine [10,42]. Wiriyathanawudhiwong et al. discovered and identified a *tolC* gene encoding an outer membrane channel involved in L-cysteine export and then overexpressed the *tolC* gene along with enhanced biosynthesis and attenuated degradation pathways to boost L-cysteine yield to 1.2 g/L [30]. Ohtsu et al. found that using thiosulfate instead of sulfate requires two ATP and four NADPH in the sulfate pathway, while only one NADPH is required in the thiosulfate pathway. Therefore, the thiosulfate pathway not only reduces the energy consumption of the assimilation pathway but also provides a more efficient sulfur source. Subsequently, the co-overexpression of glutaredoxin (NrdH), sulfite reductase (CysI), and CysK in *E. coli* facilitated biosynthesis and cellular secretion, resulting in increased yields [36]. In 2015, they identified a new *yciW* gene regulating CysB (a sulfur transcription regulatory factor) on the *E. coli* genome using the MEME and FIMO programs, and then, enhanced L-cysteine production was achieved by disrupting the *yciW* gene in *E. coli* [20]. In a recent study, an engineered *E. coli* LH16 was designed, focusing on improving the sulfur conversion rate by reducing the formation of $H_2S$, increasing the carbon flux to L-cysteine, and controlling the expression level of sulfur regulators, which ultimately increased the yield of L-cysteine. After process optimization in a 1.5 L reactor, LH16 produced 7.50 g/L of L-cysteine with a sulfur conversion of 90.11%, which is the highest sulfur conversion reported so far [14]. Thereafter, by further regulating the expression of sulfur regulators and sulfur supplements, optimizing the chassis cells based on increasing the sulfur conversion rate and the expression combinations of the genes of the synthetic pathway, and deleting the genes of the catabolic pathway, the final strain yielded 8.34 g/L L-cysteine during batch replenishment of the fermentation process [10]. On this basis, a recent study constructed recombinant *E. coli* that synergistically expresses carbon and sulfur modules for efficient synthesis of L-cysteine by combining systematic metabolic engineering with a modular balance strategy to modify the biosynthesis pathway of L-cysteine, which further increased the production of L-cysteine, resulting in a yield of recombinant *E. coli* of 11.94 g/L, which is the highest level reported so far [37].

### 2.2.2. L-Cysteine Biosynthesis in *C. glutamicum*

The metabolic pathway of L-cysteine in *C. glutamicum* is generally the same as that of *E. coli*, and L-serine is also synthesized from the glycolytic intermediate 3-phosphoglyceric acid via a three-step pathway, followed by further conversion to L-cysteine via CysE and CysK [43]. However, wild-type *C. glutamicum* is subjected to a feedback mechanism that produces almost no L-cysteine. In order to allow *C. glutamicum* to produce L-cysteine, the *cysE* gene (the gene is insensitive to feedback inhibition by L-cysteine) encoding an alteration of CysE in the *E. coli* Met256Ile mutant was introduced into *C. glutamicum*, which produces approximately 0.29 g/L of L-cysteine [38]. CysR is a transcriptional regulator that regulates sulfur metabolism in L-cysteine biosynthesis, and overexpression of the *cysR* gene in *C. glutamicum* resulted in a 2.7-fold higher intracellular sulfide concentration than that of the control strain (empty pMT-tac vector), and overexpression of the *cysE*, *cysK*, and *cysR* genes in *C. glutamicum* resulted in a 3-fold higher L-cysteine production than that of the control [27]. By disrupting the *ldh* gene encoding L-lactate dehydrogenase and the *aecD* gene encoding L-cysteine dehydrogenase, L-cysteine degradation was reduced, and the yield was increased [16]. It is also possible to enhance L-cysteine production by overexpressing potential genes encoding L-cysteine output in recombinant *C. glutamicum* [19].

In 2019, L-cysteine accumulation was achieved by knocking out the gene for CD and overexpressing the endogenous *cysE* gene. L-cysteine synthesis was then further enhanced by a variety of metabolic engineering strategies, including increasing the expression level of the key enzyme CysE, comparing several heterologous CysE to find the optimal candidate,

overexpressing CysK to increase the synthetic flux, comparing several heterologous transporter proteins to increase the L-cysteine transporter capacity, and increasing the supply of the precursor L-serine, among others. The resulting engineered strain *C. glutamicum* CYS-19 was able to accumulate 0.95 g/L L-cysteine [15]. In a recent report, Liu et al. successfully developed a bifunctional $H_2S_2$-responsive genetic circuit that is based on the concentration of $H_2S_2$ to dynamically activate and repress the expression of target genes [39]. This genetic circuit was then applied to reconstruct the L-cysteine biosynthetic network, and L-cysteine production in *C. glutamicum* reached 5.92 g/L with a sulfur conversion rate of 74.97%, which is the highest level of L-cysteine production using *C. glutamicum* reported to date [39].

2.2.3. L-Cysteine Biosynthesis in Other Bacteria

It is reported that in addition to *E. coli* and *C. glutamicum*, several other microbial bacteria have been reported to produce L-cysteine, such as *Saccharomyces cerevisiae*, *Lactococcus lactis*, *Mycobacterium tuberculosis*, and *Pseudomonas* species, which can synthesize L-cysteine from L-methionine by the reverse transsulfuration pathway (Figure 2) [44–48]. However, it was later demonstrated that L-cysteine synthesis in *Saccharomyces cerevisiae* is exclusively by L-cystathionine β-synthase (CBS; EC 4.2.1.22) and L-cystathionine γ-lyase (CGL; EC 4.4.1.1) via cystathionine [49,50]. An emerging host for fermentation to produce bio-based materials, *Pantoea ananatis*, was reported for L-cysteine production in 2017 [17]. In this study, the gene *ccdA* encoding the CD enzyme was knocked out through genetic engineering design, and the genes *cefA* and *cefB* were overexpressed. Then, when combined with the genes in the L-cysteine biosynthesis pathway, the efficient production of L-cysteine was successfully realized in *Pantoea ananatis*.

## 3. Metabolic Engineering Strategies for L-Cysteine Biosynthetic Pathway

As shown in Table 2 and Figure 3, metabolic engineering strategies to increase L-cysteine production include enhancing biosynthesis, sulfide accumulation, cell excretion, and weakening degradation.

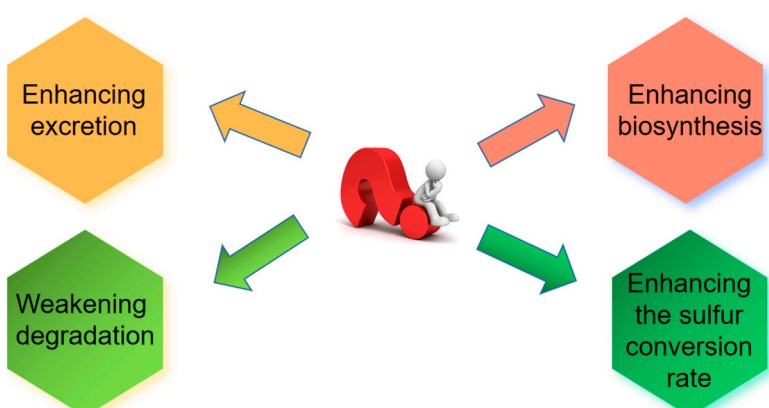

**Figure 3.** Metabolic strategies for increasing L-cysteine production.

### 3.1. Enhanced Accumulation of Precursor L-Serine

The precursor of L-cysteine is L-serine, and enhanced accumulation of L-serine is essential to increase the yield of the target amino acid L-cysteine. Firstly, the metabolic flux from 3-phosphoglycerate to L-serine can be improved, or feedback inhibition caused by high L-serine titers can be reduced, and enzymes in the L-serine biosynthesis pathway have to be regulated to be overexpressed or genetically altered [51]. Secondly, it is important to prevent the degradation of L-serine, which is not only a precursor of L-cysteine but also a precursor of glycine, L-alanine, and L-valine and is even necessary for protein synthesis, phospholipid synthesis, and C1 unit production [52,53]. Therefore, the enzymes of the relevant degradation pathways must be inhibited, or the corresponding genes must be knocked out to ensure that L-serine is converted to L-cysteine as much as possible [54–56].

Nevertheless, modulation of a single aspect does not significantly contribute to L-serine accumulation, and only trace quantities of L-serine are produced when overexpressing the *serC* and *serB* genes or modifying the gene *serA* of PGDH alone [43,57]. However, when these genes are overexpressed and co-expressed with genes that inhibit the degradation of L-serine to glycine and pyruvic acid, the production of L-serine will increase significantly [58,59]. Therefore, the combined use of multiple metabolic strategies can greatly increase the yield of target products.

### 3.2. Increase the Accumulation of Sulfides

Sulfur in L-cysteine, found in plants and bacteria, is primarily derived from sulfate. The sulfur assimilation processes of L-cysteine in *E. coli* are categorized into two main pathways: sulfate assimilation and thiosulfate assimilation pathways. In the sulphate assimilation pathway, *O*-acetyl-L-serine sulfhydrylase A catalyzes the conversion of OAS and sulfide ($S^{2-}$) into L-cysteine [14]. In the thiosulfate assimilation pathway, the entry of inorganic sulfur into the cell is achieved through the thiosulfate ABC transporter proteins, which are encoded by the genes *cysU*, *cysW*, *cysA*, and *sbp*. Thiosulfate is not only less energy intensive than sulfate but also provides a more efficient source of sulfur, making it the current optimal choice [36,60,61]. As thiosulfate is transported into the cytoplasm, it reacts with OAS via the catalytic action of *O*-acetyl-L-serine sulfhydrylase B, with the consequent formation of *S*-sulfo-L-cysteine (SSC). SSC is ultimately converted to L-cysteine by the enzymes NrdH and GrxA [62]. Thus, optimizing the thiosulfate pathway can enhance sulfide accumulation, thereby enhancing L-cysteine production.

### 3.3. Decrease the Degradation of L-Cysteine

L-cysteine degradation in *E. coli* is primarily catalyzed by the enzyme CD, which catalyzes L-cysteamine degradation to pyruvate, ammonia, and sulfide [54]. Currently, it has been determined that the degradation activity of the enzyme CD is encoded by the *aceD* gene, and disruption of the *aceD* gene, which encodes the degradation activity of the CD enzyme, has been reported to slow down the degradation of L-cysteine [16,54]. However, the concentration of L-cysteine continues to decrease during the stationary growth phase [16]. In addition, disruption of the *idh* gene encoding L-lactate dehydrogenase has been shown to prevent the formation of byproducts such as lactate [16,54].

### 3.4. Enhance the Ability of Cells to Output L-Cysteine

Advances in synthetic biology and systems metabolic engineering have provided various tools and techniques for engineering microbial strains for the production of bio-based chemicals [63]. Among these methods, efficient output of target products is a very promising method that can improve the robustness and efficiency of microbial cell factories, providing significant assistance in improving substrate absorption, overcoming metabolic inhibition, protecting cells from toxic compounds, and providing assistance for efficient fermentation production of target products [64,65].

In *E. coli*, four inner membrane transporters, namely EamB (formerly YfiK), EamA (formerly YdeD), Bcr, and CydDC, and one outer membrane transporter, namely TolC, have been confirmed to be closely related to the output of L-cysteine [4]. Overexpression of the transporter protein TolC has been shown to stimulate L-cysteine secretion, thereby attenuating the cytotoxicity of high L-cysteine concentrations and feedback inhibition [30]. Moreover, a transporter CefA was also found in *Pantoea ananatis* that is related to the output of L-cysteine [17]. Recently, it was reported that EamB, a transporter with high activity, promoted the secretion of OAS and L-cysteine in *E. coli* cells by directional evolution [12].

## 4. Challenges and New Ideas in the Biological Production of L-Cysteine

With the rapid development of synthetic biology, numerous new technologies have been created, including the new gene-editing technology CRISPR-Cpf1, CRISPR—Cas12 and HTS, DNA assembly technology, in vivo directed evolution technology, etc. [66,67]. It is

believed that the proper application of these new technologies will increase the production of L-cysteine. At the same time, new substrates, new bacteria, and new synthetic pathways to synthesize L-cysteine are still needed rather than remaining limited to the existing synthetic pathways of bacteria and in vivo metabolism utilizing glucose as the substrate and *E. coli* and *C. glutamicum* as the mainstream [21,68].

### 4.1. In Vitro Metabolic Pathways

In vitro metabolic engineering is an alternative production technology for fermentation that utilizes a cascade reaction composed of purified/semi purified enzymes for chemical production [69,70]. During production, there are no living cells in in vitro metabolic engineering, so there is no impact of toxic products on cell activity [71]. Recently, it was reported that L-cysteine was produced by in vitro metabolic engineering and that 28.2 mM L-cysteine was produced from 20 mM glucose with a molar yield of 70.5%, surpassing the greatest yield of L-cysteine produced by fermentation at present [68]. In this reaction, the genes for the enzymes needed to synthesize L-cysteine in vitro are put together in a single expression vector and co-expressed in a single strain. Additionally, the use of sodium hydrosulfide as an external sulfur source allowed for the regulation of complicated sulfur metabolism to be avoided. But there is also a problem with this reaction. Pyruvic acid must be used as a sacrificial agent, which makes this reaction not economical and restricts its industrial application.

### 4.2. Explore New Raw Materials

Currently, a large number of studies are focused on the production of L-cysteine with glucose as the substrate, whereas traditional fermentation products are also primarily concentrated in glucose, sucrose, and other sugar raw materials, limiting people's ideas.

Glycerol is one of the main by-products in biodiesel production, and according to statistics, every 10 kg of biodiesel produces approximately 1 kg of crude glycerol by-products, so utilizing glycerol would be a great asset [72,73]. With continuous research, the conversion of glycerol into high-value-added products has become more competitive [74,75]. The production of L-cysteine from *E. coli* using glycerol instead of glucose as substrate has been recently reported to have been achieved [21]. The metabolic pathway from glycerol to L-cysteine is shorter, and the carbon atom economy is more favorable compared to glucose [76,77]. Developing novel substrates for the synthesis of L-cysteine is therefore of considerable importance and value.

### 4.3. Utilization of New Technologies

The desire for efficient microbial cell factories requires constant regulation of metabolism and modulation of gene expression, which is often time-consuming [78]. The emergence of new technologies, such as CRISPR-Cpf1 gene editing and high-throughput screening (HTS), has provided a significant boost to metabolic engineering as a result of the development of synthetic biology [66,67]. In recent years, an efficient CRISPR-Cpf1 genome-editing system has been established in *C. glutamicum*, which holds promise for improving L-cysteine production [79–81]. Additionally, genetically encoded biosensors have evolved into potent instruments for high-throughput screening and real-time monitoring of metabolites in metabolic engineering [82]. On this basis, it was recently reported that biosensor-based HTS platforms are potent instruments for accelerating the development of cell factories for the production of L-cysteine and other derived compounds [83].

## 5. Conclusions

This paper reviews the progress, metabolic strategies, challenges, and future directions of L-cysteine biosynthesis. Although some progress has been made in the biosynthesis of L-cysteine on the lab scale, further research is needed to achieve industrial production. In the future, we have to dare to develop new substrates, new bacteria, and new synthetic pathways to synthesize L-cysteine. At the same time, we also need to utilize emerging

technologies such as CRISPR-Cpf1, CRISPR Cas12, HTS, DNA assembly technology, and in vivo directed evolution technology to further increase the production of biosynthetic L-cysteine.

**Author Contributions:** Writing—original draft preparation, W.L.; conceptualization and methodology, Z.Z.; writing—review and editing, project administration, and funding acquisition, D.W. All authors have read and agreed to the published version of the manuscript.

**Funding:** This work was supported by the National Key Research and Development Program of China (2022YFC2105700, 2021YFE0190800, 2021YFC2103300); the National Natural Science Foundation of China (21978027); the Fundamental Research Funds for the Central Universities (2022CDJHLW006 and 2022CDJXY-003); CAS Key Lab. of Cryogenics, TIPC (CRYO2021109); Scientific Research Foundation of State Key Lab. of Coal Mine Disaster Dynamics and Control (2011DA105287-FW202103 and 2011DA105287-ZR202002); Youth project of science and technology research program of Chongqing Education Commission of China (KJQN201900112); and Chongqing Outstanding Youth Fund (cstc2021jcyj-jqX0013).

**Institutional Review Board Statement:** Not applicable.

**Informed Consent Statement:** Not applicable.

**Data Availability Statement:** Not applicable.

**Conflicts of Interest:** The authors declare no conflict of interest.

## Abbreviations

| | | | |
|---|---|---|---|
| *Escherichia coli* | *E. coli* | Phosphoserine aminotransferase | PAST |
| *Corynebacterium glutamicum* | *C. glutamicum* | Phosphoserine phosphatase | PSP |
| *S*-Sulfo-L-cysteine | SSC | *O*-Acetyltransferase | CysE |
| 3-Phosphoglycerate dehydrogenase | PGDH | L-Cysteine synthase | CysK |
| *O*-Acetylserine | OAS | L-Cysteine desulfurase | CD |
| Gluaredoxin | NrdH | sulfite reductase | CysI |
| L-Cystathionine β-synthase | CBS | L-Cystathinine γ-lyase | CGL |

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
