# Peer review of "Recent Advances, Challenges, and Metabolic Engineering Strategies in L-Cysteine Biosynthesis"

_fermentation, doi:10.3390/fermentation9090802_

Round 1
Reviewer 1 Report
This manuscript by Li et al. discussed about the trend in L-cysteine production: past and current. Author tried to touch various metabolic engineering approached used for the efficient production of L-cysteine. However, a very few examples are highlighted and information gathered is not enough. In addition to this, the challenges and future prospectives for biological production of L-cysteine were either not discussed or discussed poorly. The authors are strongly encouraged to rewrite the whole manuscript for a cohesive and concise presentation of this work. Thus, the reviewer thought this paper need thorough revision before considering for the publication in Fermentation.
Comment 1: The title of section 2.2.1-2.2.3 is “L-cysteine biosynthesis….”. In addition to biosynthesis pathway, authors discussed about different strategy applied to increase the L-cysteine production in this section. Authors are strongly suggested to discuss only the biosynthetic pathway in this section and discuss the strategies related to modulation in section 3.
Comment 2: No detail information about the biosynthetic pathway was provided in figure. Authors are suggested to highlights/indicate the pathway and enzymes involved in L-cysteine biosynthesis pathway among E. coli, Corynebacterium and Others as discussed in text. In addition to this, authors are also encouraged to indicate the regulated enzyme and its regulation.
Comment 3: In section 3 and 4, authors highlighted the different metabolic engineering approaches/new techniques used for the enhanced production of the product. However, no information regarding to the obtained improvement was discussed. Furthermore, the metabolic strategy mentioned in Table 2 was not clearly matched with the discussion made in text. Authors are strongly suggested to address the raised issue in manuscript.
In addition to this, authors are also suggested to provide obtained yield and productivity value in Table 2.
Comment 4: Authors are suggested to discuss more properly: the challenges and future prospectives for biological production of L-cysteine in a separate section.
Comment 5: Authors show the biosynthesis pathway of L-cysteine in figure 2 and figure 3. Authors are suggested show only one figure showing the biosynthesis pathway in different organism and one separate figure regarding to different metabolic engineering approaches applied to these pathways.
Comment 6: All the figures and tables were not linked to the text. Authors are suggested to link the figures and tables to text.
Moderate editing is needed
Reviewer 2 Report
I have revised the manuscript; some comments should be considered by the authors, such as:-
1- English needs to improve; some sentences and parts of the manuscript are difficult to understand.
2- The abstract is concise; it should be enhanced by adding additional objectives and clearing the aim.
3- Compare the protease hydrolysis on protein and lysine content in a new table.
4- Make an abbreviations list at the end of the review
5- Delete the year column in Table 2
6- Unify the yield of cystine by g/L in all tables and manuscript
7- Enhance Figure 2
8- Update the review up to 2023
9- Check the outputs of all references
English needs to improve; some sentences and parts of the manuscript are difficult to understand.
Reviewer 3 Report
Please refer to the annotations in the attached PDF file

Needs some fixes to enhance a smooth flow
Round 2
Reviewer 1 Report
Authors responded the reviewer question properly and modify the manuscript accordingly. The manuscript is acceptable after minor correction.
Minor correction:
Authors only indicate the meaning of brown line and dashed line only. Authors are also suggested to indicate the meaning of orange, pink and purple line.
Minor editing
Reviewer 2 Report
The authors have carefully processed all comments. The quality of the manuscript has increased significantly. I have no further comments.
Minor editing of the English language is required.
Reviewer 3 Report
All my comments have been addressed
Requires minor fixes